# Towards responsible surveillance in preventive health data-AI research

**Sam H. A. Muller**[1,2]*, **Johannes J. M. van Delden**[1], **Ghislaine J. M. W. van Thiel**[1], **Hypermarker Consortium**¶

**1** Julius Centre for Health Sciences and Primary Care, University Medical Centre Utrecht, Utrecht University, Utrecht, The Netherlands, **2** Utrecht School of Governance, Utrecht University, Utrecht, The Netherlands

¶ Hypermarker consortium is provided in the Acknowledgments.
* smuller4@umcutrecht.nl

## Abstract

The integration of artificial intelligence (AI) into health data research promises to transform precision medicine, especially by managing complex and chronic conditions like hypertension through decision support. Yet health AI also furthers surveillance, with serious ethical and social impact. Nevertheless, surveillance in health, in particular data-AI research and innovation, is understudied. This paper provides a conceptual analysis of health data-AI surveillance using the Hypermarker research project as a case study. We trace the evolution of surveillance within medicine, public health, data-driven research, and the proliferation of digital health technologies, before examining how the development of AI technologies amplifies and transforms these existing practices. We analyse health data-AI surveillance's implications of pervasiveness and unobtrusiveness, hypercollection and function creep, hypervisibility and profiling, informational power, and the formation of a surveillant assemblage, followed by an assessment of the safeguards and measures implemented by the Hypermarker project. Our analysis exposes several key challenges for responsible surveillance practices in health data-AI research: strengthening trustworthiness through fairness and equity, ensuring accountability through transparency, and fostering public control and oversight. To this end, we recommend advancing responsible governance by implementing arrangements such as community advisory panels, independent review boards and oversight bodies, data-AI justice frameworks and dialogues, transparency dashboards and public AI portals, stewardship committees, accountability assemblies, and open oversight cycles.

### Author summary

In this study, we explore how the growing use of artificial intelligence in health research is changing the way information about people is collected,

**PLOS Digital Health**

**Data availability statement:** All relevant data are within the manuscript.

**Funding:** S.H.A.M. and G. J. M. W. received funding for this work as part of the Hypermarker project, which was funded by the European Union under Grant Agreement number 101095480 (https://doi.org/10.3030/101095480). Views and opinions expressed are however those of the authors only and do not necessarily reflect those of the European Union. Neither the European Union nor the granting authority can be held responsible for them. The funders had no role in study design, data collection and analysis, decision to publish, or preparation of the manuscript.

**Competing interests:** The authors have declared that no competing interests exist.

used, and governed. We focus on a case study that aims to improve treatment for high blood pressure and show how systems designed to support medical decisions can also create new forms of surveillance. This means that more and different kinds of information about people are gathered, often in ways they do not see or control. Such practices can affect privacy, fairness, and trust, and may deepen inequalities in healthcare. We show that these issues are not only technical or ethical, but also about how decisions are made and who is involved in making them. Building trustworthy and fair uses of artificial intelligence in health requires openness, accountability, and shared oversight. We suggest that patients, communities, and the public should have a stronger role in shaping how these technologies are developed and used.

## Introduction

The application of machine learning and advanced deep learning techniques for multimodal predictive analytics and modelling in fields like metabolomics accelerates the development of artificial intelligence (AI) systems. By bringing precision medicine closer to realization, health AI forms one of the most striking and promising frontiers in health data research [1]. Leveraging detailed insights from a wealth of data about disease, treatment, and medication effectiveness, AI-driven decision support systems [2,3] have the potential to personalize therapy and treatment for complex conditions like hypertension. These conditions not only contribute to a wide range of diseases, imposing a significant burden on patients and healthcare systems, they are also intricately linked to genetic characteristics, lifestyle, diet, and environmental factors. The interplay of these health-related factors makes effective and sustainable management and treatment particularly challenging, a problem that health AI aims to address.

While the development of AI-driven systems offers great potential for improving treatment of complex conditions with significant health implications, it also increases the scale and complexity of health data research. The associated biomarkerization highlights health research as an ongoing future-directed process that seeks to solve biomedical and public health problems through investments into biomarker research [4,5]. This further entrenches the datafication of health, disease, and care to address pressing health challenges and improve care, a process that is already well underway. Increasingly, health-related yet non-medical factors like lifestyle are incorporated into health data research, since they are pertinent to understanding and improving treatment. Moreover, the use of advanced self-learning data science techniques in systems providing predictive advice to support medical decision-making introduces a novel dimension of algorithmization in health research and care.

Accordingly, health AI research almost invariably involves forms of surveillance, defined as 'the focused, systematic and routine attention to personal details for

the purposes of influence, management, protection or direction' [6]. In fact, many ethical and social challenges already associated with health data research, such as upholding autonomy and control, privacy and data protection, trustworthiness, transparency and accountability, and fairness and equity [7,8], are closely related to risks that are linked to rising capacities for surveillance [9]. Nevertheless, although the ethical and social impact of surveillance concerning algorithmization and AI has been extensively discussed in other fields, such as policing and regulation [10,11], such discussion has been notably scarce in the context of health [12,13].

This gap is striking, since the recent shift in health data research to development of AI-driven systems seems to further proliferate and intensify the phenomenon of surveillance in health research, and by extension eventually in healthcare. So far however, a comprehensive overview and analysis of the extent of surveillance concerning its ethical and social implications in health data-AI research is lacking. Given that surveillance is increasingly becoming an integral, commonplace component of health data-AI research practices, it is crucial to determine how responsible surveillance can be furthered to realize the benefits of health AI systems for precision medicine in a legitimate and sustainable way [14].

We begin by introducing the Hypermarker research project as a case study of health data-AI research. We then trace the evolution of surveillance within medicine, public health, data-driven research, and the proliferation of digital health technologies, before examining how the development of AI technologies amplifies and transforms these existing practices. Subsequently, we analyse the implications and ethical and social issues associated with surveillance in health data-AI, and critically assess the safeguards and measures implemented in the Hypermarker project. We conclude by identifying key challenges and providing recommendations for advancing responsible surveillance practices in health data-AI research.

## Materials and methods

This paper employs a conceptual approach, using the Hypermarker research project as an illustrative case study to explore the ethical and social dimensions of surveillance in health data-AI. The analysis integrates insights from bioethics and the social sciences, in particular surveillance studies and science and technology studies (STS), to critically examine how AI-driven innovations in health research reconfigure existing practices of data collection, monitoring, and governance.

The case study of Hypermarker, a health data research project developing an AI-driven decision support system for personalization of hypertension treatment, serves as an empirical reference point for situating ethical and social issues within an ongoing research context. Relevant project documentation, public materials, and governance frameworks were reviewed to identify the safeguards and measures implemented to promote responsible research practices.

The analysis proceeds through three stages. First, it traces the evolution of surveillance practices in medicine, public health, and data-driven research, highlighting continuities and transformations introduced by AI technologies. Second, it examines the implications of these developments, highlighting several crucial ethical and social issues such as privacy, trust, transparency, and accountability. Third, it critically assesses the Hypermarker project's approach to responsible surveillance. Finally, it draws lessons for the development of ethically robust and socially responsive governance frameworks for surveillance in health data-AI research.

## Surveillance from public health to personalized prevention

The Box 1 describes the Hypermarker project, which focuses on personalized pharmacometabolomic optimization of hypertension treatment through AI-driven decision support [15]. This case description serves as a starting point to characterize the expressions of surveillance in the context of current health data-AI research.

> **Box 1.** Case description of the Hypermarker health data research project developing an AI-driven decision support system for personalization of hypertension treatment.
>
> Hypermarker is an innovative initiative focused on improving hypertension treatment through personalized pharmacometabolomics. The project aims to optimize antihypertensive therapy by tailoring treatment to each patient's unique metabolomic profile, leveraging data from large-scale cohorts across 11 European countries. A clinical decision support tool will be developed, using advanced data analytics and AI to guide healthcare providers in selecting the most effective blood pressure treatments while minimizing side effects. By integrating pharmacometabolomics, a new approach is introduced to manage high blood pressure, addressing the issue of poor medication adherence caused by adverse effects. This personalized strategy will enhance the retention of antihypertensive effects, significantly improving patient outcomes. The project seeks to meet an urgent need for more efficient use of existing hypertension therapies, enhancing systolic blood pressure control and reducing medication-related adverse events. Utilizing an advanced metabolomic platform, the project applies deep learning and AI methods to identify metabolites that predict treatment response. This data will feed into a clinical decision support model, enabling more personalized and effective care. The project will explore the role of food-drug interactions, considering how dietary factors influence treatment outcomes. Ultimately, the aim is to improve hypertension care across Europe by integrating these advanced tools into routine medical practice, ensuring more precise and effective patient treatment.

Although the development of AI-driven systems in health data research may be a relatively new phenomenon, surveillance in the context of health, medicine, and research is not. The growing reliance on health surveillance is closely linked to the rise of surveillance medicine. This reflects a shift from merely diagnosing and treating individual diseases to emphasizing early detection and prevention [16]. As a by-product, this shift problematizes normality at the population level by monitoring risk factors, potential future illnesses, and asymptomatic conditions [17]. This new medical gaze blurs the boundaries between health and disease. Surveillance medicine intensifies the focus on prevention by scrutinizing activities, behaviours, and lifestyle choices for their potential long-term health consequences, turning healthy individuals into subjects of medical concern. This is enabled by advancements in medical technology, statistical techniques, and epidemiology, allowing for the collection and analysis of vast amounts of data to track trends in populations over time. As a result, interventions increasingly focus on prevention based on risk factors, with greater emphasis placed on controlling behaviour and lifestyle choices [18,19].

Health surveillance is mostly discussed in relation to public health, where it is defined as 'the continuous, systematic collection, analysis and interpretation of health-related data needed for the planning, implementation, and evaluation of public health practice' [20]. Its scope has increasingly extended to the promotion of population-wide health in a broader context [20,21]. This is closely tied to epidemiology, which similarly involves the systematic collection and analysis of diverse data types for extensive monitoring, evaluation, and risk assessment to study health patterns and prevent disease [17,22]. In fact, much of the data used in epidemiological research originate from surveillance initially established for public health purposes [23,24]. The momentum of surveillance is further fuelled by the international reuse, sharing, and linkage of existing datasets, as well as by the growing integration of various forms of omics research, supported by advances in bioinformatics [25].

Accordingly, the intertwinement of the purposes and aims of epidemiology and public health research continues to grow [20,23], reflecting a shift away from the treatment of individual patients toward disease prevention and a move beyond the traditional boundaries of medical research [26]. Examples include influencing health policy on resource allocation, treatment decisions, and preventive consultations, where risk assessments derived from population studies are applied back to individuals, influencing the management of their risk factors and ongoing health outcomes [17].

Furthermore, surveillance has been facilitated by the rise of innovative digital technologies and tools [18,27], including health informatics such as Big Data, machine learning and deep learning AI techniques [22,28] as well as remote monitoring and tracking systems [29], such as self-monitoring, self-care, and self-surveillance systems [13,30,31]. Referred to as digital health surveillance, this system views individuals primarily as risk profiles, focusing on preventable threats and encouraging them to manage themselves as sets of health and safety risks. It places greater responsibility on patients to oversee their own health based on information generated from these profiles [19]. Examples are eHealth, the personal management of health informed by statistical analyses of individual data [18], as well as quantified self-medicine [19,29].

Furthermore, health data and surveillance are increasingly integrated into large-scale, multi-purpose information systems, creating a self-reinforcing cycle that improves and refines predictive analytics. By transcending the distinctions between individual and population, personal disease and public health, and surveillance and healthcare, the integration of population-level and self-surveillance through digital technologies has created a valuable, powerful, and productive health surveillance system [19]. This is exemplified by approaches promoting the bridging of healthcare, public health, and health research, such as real-world evidence and learning healthcare systems [32,33].

The collection, sharing, aggregation, and linkage of health-related data, especially in health data infrastructures, often take place before the specific purposes for doing so have been clearly defined, amounting to an increasingly broader emphasis on anticipating and preventing behaviour affecting health [34]. This raises critical questions about which types and sets of data are considered meaningful and valuable for this objective, how such assessments can be made, by whom, and which interests are being served [26,35]. The focus on anticipation and prevention is closely tied to the goal of personalization, or more accurately, precision medicine. Precision medicine targets specific risk categories or profiles by integrating genetic, environmental, and lifestyle factors to provide more accurate diagnoses, predict disease risks, and offer targeted treatments [36]. It departs from the traditional one-size-fits-all approach, which applies uniform treatments to all patients, and instead focuses on individual differences to optimize healthcare outcomes [31].

The rise of health data initiatives focused on developing AI-driven systems, such as decision support tools for the personalized treatment of complex conditions like hypertension, underscores the convergence of trends driving surveillance in preventive health data research and innovation. These systems exemplify how AI is reshaping healthcare by translating vast, multimodal data into actionable insights. The distinctive feature of AI lies in its enhanced ability to integrate diverse datasets, enabling their effective application in clinical practice.

Surveillance in health AI research is primarily instigated by the application of highly advanced, self-learning data science techniques that enable predictive analytics and complex multimodal modelling. Moving from public health to personalized prevention, machine learning and deep learning technologies permit the creation of (semi-)autonomous tools that can provide tailored therapeutic recommendations for conditions with complex aetiologies. This data-driven approach merges biological, behavioural, and environmental factors to craft more individualized healthcare solutions.

The integration of AI in health data research feeds back into healthcare practices and public health frameworks, influencing clinical and societal outcomes alike. However, the scale and complexity of health AI introduces new layers of health-related surveillance. The design and functionality of AI-driven decision support systems contribute to health surveillance in the form of expansive, data-intensive monitoring that is largely unnoticed yet deeply embedded within healthcare infrastructures. The routine sharing, linkage, and repurposing of diverse datasets for AI research amplify this trend, reinforcing a cycle of comprehensive health surveillance.

Accordingly, three key surveillance-oriented elements inherent in health data-AI research can be identified: (1) the sharing and linkage of new forms of omics data, along with health-affecting factors such as lifestyle, diet, and environmental influences; (2) the analysis and modelling of data using advanced self-learning algorithmic techniques; and (3) the integration of AI into tools that may (semi-)autonomously offer recommendations or prescribe treatment options for healthcare professionals. These AI-driven innovations reflect a marked shift in the field, where disease and treatment are increasingly viewed through a data-driven lens, while underscoring the pervasive nature of surveillance in modern healthcare.

## Implications and issues of surveillance in health data-AI research

The rise of digital technologies, including machine learning and deep learning for multimodal predictive analytics and modelling, has accelerated the development of health AI systems, particularly in fields like metabolomics, which in turn contribute to expanding surveillance capabilities. This development has significant implications for how health and disease, particularly complex chronic conditions, are understood, as well as how research, innovation, and care are structured around treatment as risk management.

It is important to differentiate between the descriptive, normatively agnostic use of the term surveillance in fields such as public health, epidemiology, real-world evidence, and learning healthcare systems, where the term often denotes systematic data collection and analysis for health improvement purposes, and its normative connotation as a phenomenon with broader ethical and social repercussions. Furthermore, in health data-AI research, we can differentiate between the explicit or active facilitation of surveillance, such as the development of AI algorithms for clinical decision support, and the unintentional or implicit enhancement of surveillance potential. The latter may involve partly unintended consequences or downstream effects, such as creating pathways for derivative applications that could be deployed in other health contexts or even outside the health domain entirely.

Building on this characterization of health data-AI surveillance, we examine its most important implications as well as the associated ethical and social issues. In light of these, we critically assess the safeguards and measures implemented in the Hypermarker project. These can be grouped into three main categories:

1. Technical procedures, including federated data analysis and the Findable, Accessible, Interoperable, and Reusable (FAIR) harmonization of data drawing upon the International Council for Harmonisation of Technical Requirements for Pharmaceuticals for Human Use's Guideline for Good Clinical Practice (ICH-GCP) [37].

2. Ethical and regulatory safeguards, focusing on ethics- and privacy-by-design principles, as well as data protection and privacy frameworks informed by the European Union's General Data Protection Regulation (GDPR, EU 2016/679) and Medical Device Regulation (MDR, 2017/745), the World Medical Association (WMA)'s Declaration of Helsinki [38], and the European Commission's Independent High-Level Expert Group on AI's Ethics Guidelines for Trustworthy AI [39] and Assessment List for Trustworthy AI (ALTAI) [40].

3. Patient and public involvement and engagement (PPIE), particularly in clinical trial design, health economics, and responsible governance.

## Pervasiveness and unobtrusiveness

Although regulatory and ethical frameworks increasingly emphasize the importance of explicability, the pervasive and often unobtrusive nature of surveillance in health data-AI research continues to intensify the digitalization of healthcare and disease management [41]. This digital transformation, driven by the rise of health information technology and electronic health records, increasingly positions individuals, whether patients, research participants, or donors, as passive sources of data, with minimal, let alone meaningful control over how their information is used. The integration of AI in health research further entrenches this dynamic, limiting individuals' ability as well as actual opportunities to challenge or influence the purposes for which their data are employed. The development of AI-driven algorithms and decision support systems often exacerbate the opacity of data usage, making it challenging for individuals to discern who accesses their data, for what purposes, and in which ways. This impinges upon transparency, accountability, and personal autonomy by extending surveillance beyond immediate healthcare needs, influencing treatment options, lifestyle choices, and even individuals' self-perception. For instance, AI-powered wearable devices used to monitor patients with chronic conditions, such as continuous glucose monitors for diabetes or smartwatches tracking heart rhythms, collect vast amounts of behavioural and physiological data in real time. While intended to support disease management, these tools often operate with limited

transparency regarding how data are processed, shared, or used to inform clinical decisions. Over time, such monitoring can influence not only treatment pathways but also patients' everyday choices and perceptions of health. This potentially reinforces normative ideas about 'healthy' behaviour and shifting the locus of responsibility from healthcare providers to individuals.

In Hypermarker, pharmacometabolomic profiling is designed to detect subtle biomarkers of hypertension across diverse populations. While this continuous monitoring improves early detection, it also normalizes the constant observation of biological processes. Such unobtrusiveness risks obscuring the extent to which individuals are being monitored, potentially eroding awareness of data collection boundaries. This raises ethical questions about ongoing consent, data minimization, and the extent to which patients and clinicians can meaningfully grasp the scope of AI-enabled observation embedded in everyday healthcare and research infrastructures.

The project addresses the implications of AI's pervasiveness and unobtrusiveness for transparency, accountability, and personal autonomy. Transparency is primarily operationalized through traceability and explainability measures. In selecting AI models, the project explicitly considers the trade-off between interpretability and accuracy, prioritizing interpretable models such as LASSO (Least Absolute Shrinkage and Selection Operator) and Decision Trees. More complex models are adopted only when justified by substantially improved performance. In such cases, extensive screening is conducted using explainable AI (XAI) tools, such as feature importance measures and both model-dependent and model-agnostic methods, to ensure a clear understanding of how predictions are generated. Dedicated methods are also employed to elucidate the reasoning behind more complex, so-called 'black-box' models. These explanations are made accessible to end-users in plain language and integrated into the decision-support tool's outputs. Beyond explainability, the project emphasizes transparency by adhering to established reporting standards from EQUATOR Network initiatives, including TRIPOD [42], DECIDE-AI [43], and STARD [44], which promote rigorous and transparent health AI research.

Accountability is addressed only to a limited extent, primarily through measures focused on auditability and risk management. The project promotes accountability by ensuring that code is made publicly available in open repositories, and that training data are accessible via the Hypermarker Data Catalogue and Hypermarker Federated Data Repository. These infrastructures are implemented in accordance with principles of open science and privacy-by-design, supporting transparency, reproducibility, and responsible data stewardship.

Autonomy is primarily addressed under the principle of human agency and oversight. Hypermarker designs its modelling processes to guide and support, rather than replace, human decision-making. The final predictive algorithm will be implemented as a clinical decision-support tool, accessible through a dashboard serving as the user interface for both clinicians and patients. Within this dashboard, it will be made explicitly clear that the prediction results derive from an algorithmic process. Predictions will be presented in accessible, lay terms, accompanied by detailed explanations of how they were generated. These explanations build on the project's explainable AI analyses and comply with EU in vitro diagnostic medical device regulations. Confidence intervals will be displayed and associated risks highlighted to mitigate overreliance on the system. The model functions solely as a decision-support mechanism, with final treatment decisions remaining entirely the responsibility of the clinician (human-in-command). To avoid potential confusion, no human-like simulation will be developed for end users. Instead, targeted training will be provided to ensure appropriate and informed use of the tool.

### Hypercollection and function creep

The hypercollection of multimodal data in AI-driven health systems entails accumulating vast amounts of information across diverse, often non-medical domains to detect health patterns and make probabilistic inferences [45]. Data such as lifestyle, environment, and dietary habits are increasingly aggregated with health data to improve health outcome predictions [35]. This leads to the medicalization of health-related data that are not medical, let alone strictly health-related or health-relevant. This introduces risks of function creep, whereby data initially collected for specific health-related purposes are later repurposed for unrelated or broader objectives. Such repurposing can result in privacy overreach – that

is, the excessive collection, use, or sharing of personal data beyond what individuals initially consented to or reasonably expected, thereby eroding trust and compromising informational self-determination [46]. As multiple stakeholders, including governments, insurers, and commercial entities, seek access to these data, the likelihood of misuse increases, at the risk of compromising privacy and eroding trust of those contributing and providing the data. Wearable devices and health apps, at first sight designed to deliver personalized health insights [47], further reinforce these issues by shifting surveillance to patients themselves and promoting self-regulation through data feedback loops that encourage conformity to health norms [9,13,30]. As a result, individuals increasingly become both the monitored subjects and active participants in their own surveillance, encouraged to continuously track and interpret their health data. While this data-driven approach promises more personalized care, it also risks depersonalizing healthcare by prioritizing algorithmic profiles over lived experiences, clinical judgement, and relational aspects of care.

Hypermarker exemplifies hypercollection, the aggregation of vast, heterogeneous datasets including biological samples, clinical trial data, and multi-omic information. While these data are crucial for training robust predictive models, their richness increases the likelihood of function creep, where data are repurposed for new analyses or commercial use beyond the project's original intent. For instance, pharmacometabolomic data collected for hypertension research could later be used for unrelated biomarker discovery without renewed consent. The very structure that enables large-scale integration also amplifies the ethical tension between scientific value and data sovereignty.

The project addresses privacy challenges arising from hypercollection and function creep by focusing on privacy standards, data quality and integrity, secure access, and stakeholder participation. While trust is not explicitly addressed, privacy protection is operationalized through multiple technical and procedural safeguards. All data collection and analytical activities are conducted in compliance with recognized privacy standards to ensure the quality and integrity of data. Data usage policies are communicated to users, and informed consent is obtained for the collection and processing of personal information. In addition, robust procedures are implemented to secure data access in accordance with GDPR principles. For the final decision-support tool, the FAIR Digital Twinning approach incorporates encryption at the source and a multi-level privacy protection architecture. Only the subsets of data necessary for a specific research question are used, in alignment with the user's consent. A digital 'twin' of a real-world individual is created for analysis, while end-users can access only the analytical outcomes within a safe, secure, and FAIR environment, promoting the ethical use and responsible reuse of patient data. Patients, clinicians, and other relevant stakeholders are involved in the design and development of the clinical decision-support tool. Through co-creation, the project aims to embed values such as privacy into the resulting technologies and governance processes.

## Hypervisibility and profiling

AI-driven health data systems frequently employ profiling to stratify risks and personalize care [48,49]. However, this often-covert practice significantly shapes patient experiences, influencing treatment options, access to care interventions, and interactions with care providers. Profiling may also reinforce existing biases, particularly against marginalized or vulnerable populations, by categorizing individuals and groups based on socioeconomic status or demographic factors, exacerbating healthcare disparities [8,12]. This heightened hypervisibility can lead to surveillance intensifying on specific groups, often without tangible benefits and at the risk of stigmatization and discrimination. The reliance on algorithmic categorizations thus threatens trust, transparency, and accountability, as both patients and healthcare professionals remain unaware of the criteria informing care decision support. This impedes their ability to contest decisions and potentially unfair or biased outcomes. Ultimately, profiling risks entrenching and reinforcing systemic healthcare inequities, undermining efforts toward more equitable health outcomes.

Hypermarker's predictive algorithms stratify patients according to risk of hypertension and treatment response, potentially reproducing or amplifying existing inequities. If training data underrepresent certain demographic or socioeconomic groups, the resulting models may produce skewed risk profiles that render already disadvantaged populations

hypervisible as 'high risk'. This could reinforce stigma or justify differential treatment pathways. For example, minority or low-income groups disproportionately affected by hypertension may face intensified monitoring or exclusion from tailored interventions.

The Hypermarker project addresses the impact of profiling and hypervisibility, manifested through bias, unfairness, inequities, and potential stigmatisation, by emphasising diversity, non-discrimination, mitigation of unfair bias, and stakeholder participation. Before addressing potential biases in AI modelling, inherited biases in data that underpin these models but may not fully represent population diversity are considered. To mitigate such effects, the project commits to transparent reporting and communication of findings, guided by established reporting standards such as STROBE [50]. Balanced sampling strategies and metadata considerations are employed to address potential biases in training data and to promote fairness across dimensions such as gender, ethnicity, and disability. Regarding gender, the derivation cohort is drawn from multiple countries with gender parity in sampling. Trial recruitment includes stratification of the primary outcome by gender, and algorithm development explicitly treats gender as a key covariate. Through participatory processes, the project aims to embed non-discrimination and service to the public good into the development and governance of health AI.

## Informational power

The accumulation of health data by private entities, such as insurers, tech firms, and data brokers, creates significant informational power that enables these actors to influence premiums, healthcare costs, access to care, and eligibility based on risk profiles. This dynamic disproportionately impacts high-risk individuals, increasing their financial burdens and restricting access to care. Such informational power extends beyond traditional healthcare providers to commercial entities controlling health data algorithms, facilitating surveillance and control that reach well beyond patient care [35]. The integration of health and non-health data underscores the blurring of boundaries between medical and societal oversight, further complicating accountability. As non-healthcare entities exert growing control over health data usage, exploitative practices may emerge, undermining both public trust and equity in healthcare.

The data infrastructures underpinning Hypermarker exemplify the informational power characteristic of contemporary surveillance. The capacity to collect, integrate, and infer from multi-modal health data positions AI systems as powerful epistemic agents, shaping not only medical knowledge but also the allocation of resources and the definition of health risks. In Hypermarker, predictive insights could influence clinical decisions or policy priorities, thus redistributing epistemic authority away from patients and practitioners toward algorithmic systems and data managers.

Beyond accountability (as addressed above), the effect of informational power on public trust and equity is not directly addressed in Hypermarker.

## The surveillant assemblage

Surveillance in health data-AI research highlights the interconnected network of health data research, development of AI-driven systems, and implementation for decision support in care that extend beyond individual privacy and autonomy concerns to broader social and political implications [28]. This surveillant assemblage entails health data systems that increasingly operate within a framework dominated by algorithmization [51], where algorithmic processes shape healthcare decisions, policies, access, as well as societal norms. This model, often driven by private-sector data commodification [35], amplifies asymmetries between citizens and corporations, producing a healthcare system akin to a panoptic structure where surveillance is central [17,52]. In this system, patients' behaviours may be subtly influenced through algorithmic nudges and monitoring, often without mechanisms for contesting or opting out, turning privacy into a collective societal concern under both public and private sector surveillance. As health AI systems evolve, robust policy and governance measures are crucial to mitigate their potential harm, ensure responsible and transparent use of AI, safeguard patient autonomy, and promote equity.

Hypermarker operates within a surveillant assemblage, a distributed network of actors, technologies, and institutions that collectively produce and manage health data. This assemblage includes hospitals, research institutions, data repositories, AI developers, and governance bodies. Each contributes to the continuous flow, analysis, and interpretation of personal health information. While this networked model enables collaborative innovation, it also diffuses accountability, since no single actor fully controls how data circulate or are repurposed. The complexity of distributed surveillance raises persistent questions about how to maintain ethical coherence across diverse sites of control and responsibility.

Beyond privacy, autonomy, and transparency (as discussed above), the surveillant assemblage's effects are not addressed in the Hypermarker project.

## Key challenges for responsible surveillance practices in health AI research

The ethical and social issues associated with the implications of surveillance, together with our assessment of safeguards and measures implemented in the Hypermarker project, highlights key challenges for advancing responsible surveillance practices in health data-AI research. [39] First and foremost, there appears a need to build trustworthiness regarding concerns about upholding social value, more precisely equity and fairness, in the development of health AI systems. This particularly applies to decision support for complex chronic conditions, for which management and treatment are linked to a variety of factors, including non-medical, health-related ones such as lifestyle, socioeconomic, and demographic status that lie far from the medical purview. Second, fostering effective accountability and greater transparency proves to be warranted for preventive data research and innovation, given the complexity and opacity of digital technological developments involving large-scale data usage and advanced data science techniques. Third, strengthening public control and oversight seems necessary since individual autonomy such as in the form of consent, coupled with local ethical oversight and review, appears rather ill-suited to the wider societal impact of health data-AI surveillance.

Therefore, health data-AI surveillance should not be taken for granted or simply left to existing regulation and guidelines regarding the use of AI or large-scale reuse, sharing, and/or linkage of health-related data. In fact, we contend that this is what makes surveillance a problematic phenomenon: it largely falls outside of the purview of most professionals and experts working on the development of AI systems within preventive health data research, yet its implications are also quite hard to pin down from a confined ethico-legal perspective. Moreover, the critical literature on surveillance does an ill job at providing practical recommendations to improve existing governance strategies, policies, and measures.

## Trustworthiness as prerequisite for fairness and equity

Trustworthiness is central to establishing a fair and just healthcare system, particularly as emerging technologies such as AI assume an increasingly prominent role. Yet trust is neither static nor one-dimensional: it is continuously reshaped by institutional arrangements that influence whether stakeholders place trust in health AI systems or, conversely, develop mistrust [53]. In this context, trustworthiness must be warranted through governance that upholds fair and equitable research and innovation practices that are responsive to people's needs, concerns, and expectations.

A key challenge in fostering trustworthiness lies in the need for symmetrical and reflexive approaches, understanding trust not merely as something to be earned by experts, but as relational, shaped by the concerns, values, and lived experiences of diverse stakeholder communities and publics [54]. For example, in health data-AI research, communities may hesitate to trust predictive algorithms for early disease detection if their past experiences with healthcare systems involved bias, opacity, or lack of control over their data. Addressing this requires engaging with such communities not just to explain the technology, but to understand their perspectives and incorporate their feedback into development, implementation, and governance processes [53].

Sustaining trust requires identifying and establishing the conditions that enable repeated and reproducible trust in the development and implementation of health AI systems. This is not simply an outcome of technological advancement. Beyond individual interactions, trustworthiness concerns the ongoing reliability of systems and practices, processes that

must align with the interests and expectations of those affected [55]. The design and development of health AI systems should therefore ensure that these technologies serve the collective interests of the communities they are intended to support, while upholding individuals' rights, dignity, and self-determination. This demands research and governance practices that are responsive to public expectations of fairness, transparency, and respect for human rights.

Trustworthiness thus plays a crucial role in advancing equity in health data-AI. For these systems to be fair and socially acceptable, they must be developed and implemented in ways that actively recognize and address structural inequalities. Here, trustworthiness entails more than technical reliability: it requires a commitment to inclusivity, transparency, and responsiveness to the needs of historically marginalized or underserved groups [8,56]. This is particularly critical for addressing biases in AI systems, which can skew healthcare outcomes and disproportionately affect marginalized populations [57]. For example, an AI model trained primarily on data from affluent, urban populations may produce biased outcomes for rural or minority communities. Without deliberate efforts to ensure data diversity, meaningful stakeholder engagement, and accountability throughout development and implementation, such systems risk reinforcing disparities rather than alleviating them.

Preventing these biases requires embedding fairness throughout design, development, and deployment. Concrete strategies include systematically auditing datasets for bias, ensuring that ethnicity and other demographic variables are accurately recorded and represented, and meaningfully including ethnic minority groups in clinical trials that generate training data [57]. Such measures help ensure that AI systems promote equitable access to healthcare and support fair treatment for all individuals, regardless of their background or identity. Equitable access also demands that decision-support systems for complex or chronic conditions do not inadvertently disadvantage certain groups.

## Establishing accountability through transparency

The integration of AI into healthcare has the potential to transform care delivery, offering benefits such as improved diagnosis, personalized treatment, and more efficient health services. Yet these advances are accompanied by significant risks, particularly concerning privacy, data security, and confidentiality. While surveillance in health AI research enables the large-scale collection of data that drives innovation, it also raises critical concerns about the protection of sensitive information and the potential for misuse or exploitation.

Given these risks, fostering transparency and ensuring appropriate accountability are paramount, particularly in light of the often opaque surveillance practices embedded in digital health technologies. Surveillance in this context typically involves the continuous, large-scale collection and analysis of personal health data, often without individuals being fully aware of the extent or implications of such monitoring. The complexity and opacity of health AI systems, especially those relying on vast datasets and advanced techniques such as predictive analytics and behavioural modelling, further complicate this need. For example, remote patient monitoring tools may capture not only clinically relevant data, such as blood pressure, but also behavioural information like daily activity or sleep patterns. These data can be silently fed into algorithms that predict health risks. As health data-AI increasingly drives preventive research and innovation, governance must explicitly address the expansion of data use beyond its original purpose and the normalization of constant monitoring through ostensibly benign health technologies.

Without clear communication about what data are collected, how they are processed, and for what purposes they are used, individuals cannot properly assess the consequences of their participation [58]. Transparency thus becomes essential to ensure that healthcare institutions and private entities deploying AI systems can be held accountable for their impacts on privacy, autonomy, and the broader well-being of affected communities [35]. It is therefore a crucial mechanism for revealing and scrutinizing how surveillance operates and for empowering individuals to make informed decisions about their participation.

Making these practices visible is essential for safeguarding privacy, preventing discrimination, and ensuring that digital health innovations genuinely serve the public good. The successful integration of health AI systems depends on balancing

their potential benefits with the specific risks posed by surveillance practices. This requires a strong commitment to transparency, not only to ensure system effectiveness, but to clarify responsibilities for data collection, processing, and secondary use [59], all of which must remain open to public scrutiny.

These challenges underscore the importance of broad societal involvement in shaping the ethical frameworks and governance of health data-AI research. Regulatory initiatives have already laid important foundations by emphasizing transparency, accountability, and human oversight. Yet the ongoing development and implementation of AI systems must continue to be guided by collective agreements that genuinely reflect social values and serve the public good [60–62]. In this sense, accountability enables society to exert meaningful influence over technological development, ensuring that health AI upholds ethical principles and promotes public interests.

### Public control and oversight

Strenghtening public control and oversight underscores the need for meaningful influence in health AI governance. These aims are closely interlinked, particularly when balancing individual autonomy with broader societal involvement in data use for health AI development. While autonomy remains a critical ethical concern, the emphasis should extend beyond personal freedoms to include public oversight. Such oversight helps ensure social licence, fostering responsiveness to the needs of stakeholders whose voices might otherwise be marginalized.

Public structures that facilitate interaction between scientists, users, and policymakers are essential for achieving this balance. These structures should enable meaningful dialogue among a wide range of stakeholders, including affected communities, publics, scientific experts, policy-makers, and political decision-makers, through platforms that move beyond a narrow focus on individual autonomy to address collective demands, societal values and expectations, and public interests. Micro-level forms of control, such as specific informed consent as well as local ethical oversight and review, are often ill-suited to addressing the wider social impacts of surveillance practices reinforced by health data-AI research.

Such surveillance can produce subtle but far-reaching effects. Predictive algorithms might categorise individuals as 'at risk' of certain diseases based on behavioural or genetic profiles, leading to potential discrimination by insurers or employers. Similarly, remote monitoring devices designed for preventive care may continuously track lifestyle behaviours, such as physical activity, diet, or sleep, without individuals fully understanding how this information might be used for profiling or rationing healthcare. Here, the harm extends beyond breaches of privacy: it reshapes social relations, reinforces health inequities, and alters access to essential services.

Addressing these systemic effects demands oversight mechanisms capable of responding to the collective implications of surveillance. Recognizing that these effects are shaped by political as well as technical decisions, deliberation among stakeholders must critically engage with the broader social and political contexts in which the development and implementation of these systems are situated. Public oversight mechanisms are needed to ensure that health data use remains publicly acceptable and aligned with stakeholder values. Such structures enable ongoing scrutiny and collective decision-making, ensuring that data practices for health AI development reflect broader societal priorities rather than institutional or commercial interests alone. They provide both oversight and practical mechanisms that foster public control and protect the interests of communities and publics [55].

### Recommendations for governance of health data-AI surveillance

Building on the relational understanding of trust, community advisory panels can help translate principles for trustworthiness into practice. To counteract hypercollection and function creep, such participatory structures enable sustained deliberation between researchers, developers, and community representatives, ensuring that public concerns shape decisions around data use, algorithmic design, and the visibility of data collection. For instance, research and innovation initiatives could employ community advisory panels to co-design transparency and communication strategies for AI diagnostic tools, demonstrating how trust can be built through visible and repeatable engagement.

Another way to operationalize reproducible trust is through independent review boards, such as ethics or equity boards, that conduct ongoing reviews of fairness and bias in AI systems. These boards can integrate algorithmic impact assessments throughout research lifecycles to monitor and address emerging equity risks, ensuring that trustworthiness remains a continuous and verifiable process rather than a static attribute. By doing so, they counteract the informational power held by data managers and commercial actors, redistributing epistemic authority back to patients and practitioners.

Profiling and hypervisibility require targeted governance mechanisms to mitigate bias, inequity, and stigmatization. Developers and researchers can adopt a data-AI justice framework that embeds fairness and inclusivity at the design stage. This involves systematic auditing of datasets for demographic bias, validation with affected groups, and transparent documentation of data provenance. In addition, initiatives could pilot data-AI dialogues with underrepresented communities to co-design metadata and ensure that local health realities are reflected in data infrastructures, and that data are not repurposed beyond the scope of consent.

Accessible transparency dashboards that report AI performance across demographic groups can sustain public confidence, making algorithmic risk stratifications and real-time AI monitoring visible and contestable, restoring participant awareness and agency in contexts where surveillance is otherwise invisible. Together, these measures help ensure that fairness and equity are not abstract aspirations but lived realities within evolving health AI systems. Transparency can be further strengthened through public AI portals, digital platforms that allow people to see what data are collected, how algorithms function, and who is accountable for their outcomes. In light of the unobtrusive and pervasive monitoring inherent in AI-driven health systems, such portals could explain the purpose, data sources, and decision logic behind AI systems in health services, provide simplified descriptions of AI models, publish regular performance audits, and include accessible channels for questions or complaints.

Translating accountability through transparency into practice requires visible and sustainable accountability mechanisms. One effective approach is the establishment of stewardship committees comprising researchers, patient representatives, and civil society members who review data-sharing practices, secondary data use, and algorithmic development decisions across the network, ensuring coherence in a distributed system. These committees can conduct multi-stakeholder reviews of research proposals, ensuring that innovation proceeds within clearly defined and publicly legitimate boundaries.

Participatory models such as accountability assemblies or juries, which deliberate on the ethical and social implications of health data-AI surveillance, can also strengthen institutional accountability [63]. These forums provide structured opportunities for communities and publics to question data practices, review audit results, and co-create principles for acceptable data use, thereby directly addressing the risks posed by stratified surveillance, fostering informed public influence over AI governance.

Public control on the surveillant assemblage, where data circulate across distributed actors and institutions, can be further institutionalized through independent oversight bodies that combine professional expertise and lived experience, moving beyond consultation towards co-decision-making and giving communities real influence over how data are used and algorithms validated in alignment with public values. To complement consent-based mechanisms and local review, wider citizen assemblies can be convened to deliberate key governance questions, such as data sharing with industry. By integrating public reasoning into technical decision-making, these assemblies enable publics to weigh societal trade-offs collectively and ensure that data and AI innovation evolves within democratically negotiated boundaries. In addition, open oversight cycles, consisting of periodic public reviews in which oversight bodies publish plain-language reports on how data are used, shared, and governed, can further reinforce accountability.

Ultimately, effective public control depends on layered, participatory structures that connect everyday governance with strategic oversight. Combining community advisory panels, independent review boards and oversight bodies, data-AI justice frameworks and dialogues, transparency dashboards and public AI portals, stewardship committees, accountability assemblies, and open oversight cycles enables public influence to operate across everyday data-AI practices, strategic

oversight, and broader policy decisions. These structures ensure that trust, equity, and accountability are co-produced in practice, rather than imposed externally, and that the ethical challenges of pervasive, hypercollected, and algorithmically mediated health surveillance are directly addressed.

## Conclusion

The surveillance induced by health data-AI research entails implications of pervasiveness and unobtrusiveness, hypercollection and function creep, hypervisibility and profiling, informational power, and the formation of a surveillant assemblage. While the safeguards and measures implemented by the Hypermarker project address many of the ethical and social issues arising from these dynamics, they also expose several persistent challenges for responsible surveillance practices in health data-AI research. Strengthening trustworthiness through fairness and equity, ensuring accountability through transparency, and fostering public control and oversight emerge as key governance challenges. To this end, mechanisms such as community advisory panels, independent review boards and oversight bodies, data-AI justice frameworks and dialogues, transparency dashboards and public AI portals, stewardship committees, accountability assemblies, and open oversight cycles play an essential role in advancing responsible surveillance practices in health data-AI research.

## Acknowledgments

This work was conducted as part of the Hypermarker consortium. The authors gratefully acknowledge the support and collaboration provided by the consortium partners throughout the course of this research.

## Author contributions

**Conceptualization:** Sam H. A. Muller, Johannes J. M. van Delden.

**Funding acquisition:** Sam H. A. Muller, Ghislaine J. M. W. van Thiel.

**Project administration:** Johannes J. M. van Delden.

**Supervision:** Johannes J. M. van Delden, Ghislaine J. M. W. van Thiel.

**Writing – original draft:** Sam H. A. Muller.

**Writing – review & editing:** Johannes J. M. van Delden, Ghislaine J. M. W. van Thiel.

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
