## [Decision Letter · Decision Letter 0]

4 Sep 2025

Response to Reviewers
Revised Manuscript with Track Changes
Manuscript
**Journal Requirements:**

2. Please provide a/amend your detailed Financial Disclosure statement. This is published with the article. It must therefore be completed in full sentences and contain the exact wording you wish to be published.

**Please only choose the relevant sentences from below**

a. Please clarify all sources of funding (financial or material support) for your study. List the grants (with grant number) or organizations (with url) that supported your study, including funding received from your institution.

b. State the initials, alongside each funding source, of each author to receive each grant.

c. State what role the funders took in the study. If the funders had no role in your study, please state: “The funders had no role in study design, data collection and analysis, decision to publish, or preparation of the manuscript.”

d. If any authors received a salary from any of your funders, please state which authors and which funders.

**Additional Editor Comments (if provided):**

**Reviewers' Comments:**

**Comments to the Author**

1. Does this manuscript meet PLOS Digital Health’s publication criteria?

Reviewer #1: Yes

Reviewer #2: Partly

2. Has the statistical analysis been performed appropriately and rigorously?

Reviewer #1: N/A

Reviewer #2: N/A

3. Have the authors made all data underlying the findings in their manuscript fully available (please refer to the Data Availability Statement at the start of the manuscript PDF file)?

Reviewer #1: No

Reviewer #2: Yes

4. Is the manuscript presented in an intelligible fashion and written in standard English?

Reviewer #1: Yes

Reviewer #2: Yes

Reviewer #1: The paper is coherent and well-structured. The authors effectively build their case by establishing the historical context of surveillance in medicine and then demonstrating how AI technologies amplify and transform these existing practices. In my opinion, the paper makes a significant contribution to the field by moving beyond a purely critical stance to offer a concrete and actionable framework for governance, thereby addressing a noted gap in the critical surveillance literature. Despite its considerable strengths, the paper's argument is not without weaknesses and important nuances that must be considered. The most significant of these is its use of the Hypermarker project as its central case study. The analysis is selective, highlighting the project as an exemplar of surveillance risks while failing to engage with the project's own explicit and documented commitments to ethical safeguards such as "privacy by design" and stakeholder co-creation. This omission makes the use of the case study appear more as a convenient illustration than as a balanced piece of evidence. It should be noted that the paper's heavy reliance on European scholarship and its grounding in a policy context shaped by regulations like the GDPR may limit the direct applicability of its specific governance recommendations in other legal and political environments, such as the more sector-specific privacy landscape of the United States.

Reviewer #2: Alright, here’s the thing. This paper takes on a big, timely, kinda scary topic—health surveillance in the age of AI. You do a nice job walking us through the history of surveillance and then setting the stage with today’s personalized medicine and hyper-data-collection mess. The bit about intentional vs unintentional surveillance? Super useful. And the way you point out issues like hypercollection, profiling, and power concentration—yeah, spot on. Also, kudos for the framework on trustworthiness, transparency, and collective oversight. That’s gold.

But. (You knew there was a “but,” right?)

This still reads more like a polished perspective piece than a research article. Which is fine—except you’ve slotted it under “Research Article.” So let’s talk tweaks.

Major Revisions

1. Bring Hypermarker into the spotlight.

You dangle this fascinating Hypermarker project as a case study… and then barely touch it again. It sits mostly in a box like an afterthought. That’s a missed opportunity. The theoretical discussion is good, but readers need to see those concepts in action.

So: when you talk about hypercollection and function creep, connect it directly—what happens if Hypermarker’s pharmacometabolomic data gets reused down the line? When you dig into profiling and hypervisibility, who exactly might be most at risk if Hypermarker’s outputs lean into known hypertension disparities? And those governance recommendations—what would collective oversight actually look like in this specific consortium? Pull Hypermarker through the narrative instead of leaving it parked on the side.

2. Clarify the research “method.”

Right now, the structure is a bit confusing. You call it a research article, but it doesn’t walk or talk like one—there’s no Methods section. Readers will appreciate a short, no-frills “Methods” chunk right after the intro. Something like: this is a conceptual analysis using Hypermarker as an illustrative case, supported by literature from bioethics, surveillance studies, and health policy. Boom. Clear.

3. Make the recommendations bite-sized and practical.

The trust/oversight/accountability trio is strong in theory, but at times it feels like you’re waving from 30,000 feet. Readers—especially policymakers and researchers—need to know: how does this translate on the ground? If you say “participatory governance structures,” don’t stop there. Spell out one or two models. Maybe a citizen advisory board. Maybe a data trust with teeth. Maybe structured co-design workshops where patients and communities actually get a voice. Concrete examples will make this much more actionable.

Minor Revisions

Trim a bit. Some sentences are long and winding. Breaking them up will keep readers from losing the thread. You don’t have to dumb it down—just give the brain a few more pauses.

**Do you want your identity to be public for this peer review?** For information about this choice, including consent withdrawal, please see our Privacy Policy

Reviewer #1: **Yes: ** Ajai Sehgal

Reviewer #2: **Yes: ** MD TOUFIQ HASSAN SHAWON

**Figure resubmission:**

**Reproducibility:** To enhance the reproducibility of your results, we recommend that authors of applicable studies deposit laboratory protocols in protocols.io, where a protocol can be assigned its own identifier (DOI) such that it can be cited independently in the future. Additionally, PLOS ONE offers an option to publish peer-reviewed clinical study protocols. Read more information on sharing protocols at https://plos.org/protocols?utm_medium=editorial-email&utm_source=authorletters&utm_campaign=protocols

---

## [Decision Letter · Decision Letter 1]

4 Dec 2025

Towards responsible surveillance in preventive health data-AI research

PDIG-D-25-00594R1

Dear dr. Muller,

We are pleased to inform you that your manuscript 'Towards responsible surveillance in preventive health data-AI research' has been provisionally accepted for publication in PLOS Digital Health.

Best regards,

Po-Chih Kuo, Ph. D.

Section Editor

PLOS Digital Health

**Additional Editor Comments (if provided):**

**Reviewer Comments (if any, and for reference):**

Reviewer's Responses to Questions

**Comments to the Author**

Reviewer #1: All comments have been addressed

publication criteria?

Reviewer #1: Yes

3. Has the statistical analysis been performed appropriately and rigorously?

Reviewer #1: N/A

4. Have the authors made all data underlying the findings in their manuscript fully available (please refer to the Data Availability Statement at the start of the manuscript PDF file)?

Reviewer #1: Yes

5. Is the manuscript presented in an intelligible fashion and written in standard English?

Reviewer #1: Yes

Reviewer #1: The authors have substantially improved the paper and addressed all comments. I recommend its acceptance.

**Do you want your identity to be public for this peer review?** For information about this choice, including consent withdrawal, please see our Privacy Policy

Reviewer #1: **Yes: ** Ajai Sehgal
